# Synthesis of MIL-Modified Fe_3_O_4_ Magnetic Nanoparticles for Enhancing Uptake and Efficiency of Temozolomide in Glioblastoma Treatment

**DOI:** 10.3390/ijms23052874

**Published:** 2022-03-06

**Authors:** Luca Pulvirenti, Francesca Monforte, Francesca Lo Presti, Giovanni Li Volti, Giuseppe Carota, Fulvia Sinatra, Corrado Bongiorno, Giovanni Mannino, Maria Teresa Cambria, Guglielmo Guido Condorelli

**Affiliations:** 1Dipartimento di Scienze Chimiche, Università di Catania, Viale Andrea Doria 6, 95125 Catania, Italy; luca.pulvirenti@phd.unict.it (L.P.); marzia.monforte@gmail.com (F.M.); chiccalopresti@gmail.com (F.L.P.); 2Dipartimento di Scienze Biomediche e Biotecnologiche, Sezione di Biochimica, Università di Catania, Via S. Sofia 92, 95125 Catania, Italy; livolti@unict.it (G.L.V.); giuseppe-carota@outlook.it (G.C.); sinatra@unict.it (F.S.); 3CNR-IMM, Strada VIII no. 5 Zona Industriale, 95121 Catania, Italy; corrado.bongiorno@imm.cnr.it (C.B.); giovanni.mannino@imm.cnr.it (G.M.)

**Keywords:** metal organic frameworks, Fe_3_O_4_, magnetic nanoparticles, glioblastoma, temozolomide

## Abstract

A nanometric hybrid system consisting of a Fe_3_O_4_ magnetic nanoparticles modified through the growth of Fe-based Metal-organic frameworks of the MIL (Materials Institute Lavoiser) was developed. The obtained system retains both the nanometer dimensions and the magnetic properties of the Fe_3_O_4_ nanoparticles and possesses increased the loading capability due to the highly porous Fe-MIL. It was tested to load, carry and release temozolomide (TMZ) for the treatment of glioblastoma multiforme one of the most aggressive and deadly human cancers. The chemical characterization of the hybrid system was performed through various complementary techniques: X-ray-diffraction, thermogravimetric analysis, FT-IR and X-ray photoelectron spectroscopies. The nanomaterial showed low toxicity and an increased adsorption capacity compared to bare Fe_3_O_4_ magnetic nanoparticles (MNPs). It can load about 12 mg/g of TMZ and carry the drug into A172 cells without degradation. Our experimental data confirm that, after 48 h of treatment, the TMZ-loaded hybrid nanoparticles (15 and 20 μg/mL) suppressed human glioblastoma cell viability much more effectively than the free drug. Finally, we found that the internalization of the MIL-modified system is more evident than bare MNPs at all the used concentrations both in the cytoplasm and in the nucleus suggesting that it can be capable of overcoming the blood-brain barrier and targeting brain tumors. In conclusion, these results indicate that this combined nanoparticle represents a highly promising drug delivery system for TMZ targeting into cancer cells.

## 1. Introduction

Glioblastoma multiforme (GBM) is one of the most common, deadly and difficult to treat brain tumors [1]. Surgical removal of the tumor, followed by radiotherapy and the oral administration of temozolomide (TMZ), is the current pharmacological treatment, but this regimen only improves the patient’s overall survival [2]. The current median survival (MS) for patients with glioblastoma is about 15 months; the average five-year survival rate is less than 5% [3,4]. TMZ is the first-choice chemotherapeutic agent in GBM which, at physiological pH, is spontaneously hydrolyzed into the highly unstable compound 5-(3-methyltria-zen-1-yl) imidazole-4-carboxamide (MTIC), which rapidly degrades into 5-aminoimidazole-4-carboxamide (AIC) and methyldiazonium ion, a DNA alkylating species [5]. Compared to TMZ, MTIC has a poor penetration of the blood brain barrier and a reduced cellular absorption. Therefore, the accumulation of therapeutically effective amounts of MTIC at the tumor site depends on the stability of TMZ. However, due to the rapid elimination rate and the short half-life of approximately two hours, most of the oral administered TMZ does not reach the tumor [6]. In fact, only about 20% of the administered dose of TMZ is generally detectable in cerebrospinal fluid. High doses of TMZ must therefore be administered repeatedly to achieve an effective anticancer effect. However, this leads to undesirable effects such as myelosuppression [7,8]. Furthermore, the efficacy of walnut septum extract (*Juglans regia* L.) as co-adjuvant treatment in cancer therapy against glioblastoma has been investigated [9]. Therefore, an important issue for GBM treatment is the development of a suitable carrier to protect and deliver a specific drug such as TMZ. In this regard nanostructured systems such as liposomes, polymeric and functionalized inorganic nanoparticles have shown promising results in the delivery of active drugs to the brain [10,11,12,13] and, in particular, organic functionalized Fe_3_O_4_ nanoparticles have attracted increasing attention [14,15] because of the combination of the magnetic properties and biocompatibility of the iron core with the versatility of the functionalization shells [16,17]. Recently, Metal-Organic Frameworks (MOFs) coordination polymers consisting of metal nodes and polydentate organic linkers organized in open porous structures [18,19] and, among them, MILs (Materials Institute Lavoiser) an MOF subclass constituted by trivalent transition metals and bi- or tri-carboxylic ligands, showed great potentiality for biomedical applications either as pure crystals or as composite materials, because of their biocompatibility and capability of loading molecules in their porous structure [20,21,22,23,24]. Moreover, some paper reported on the possibility of combining magnetic nanoparticles with MILs structures either as composite materials made of MILs crystals decorated or loaded with magnetic Fe_3_O_4_ nanoparticles [25,26,27,28] or core-shell systems in which an Fe_3_O_4_ core is covered with a MIL shell [29,30,31,32,33,34,35]. In both cases typically particles sizes ranged from 200–500 nm. These systems were successfully used for applications ranging from environmental remediation to catalysis and biosensing. However, the large size of these composites limited their use for drug-delivery applications, especially for brain cancer therapy.

In this paper, a synthetic route was developed to modify Fe_3_O_4_ nanoparticles with MIL-based structures formed by Fe^3+^ and 2-aminoterephthalate avoiding the growth of large MIL crystals. Fe MILs obtained using terephthalate ligands usually exist in various stereoisomers (the most common are MIL-101, 88B and 53) with the general formula [Fe_3_(O)X-(Solv)_2_(C_8_H_5_NO_4_)_3_] [20,36,37], (X = Cl^−^ or OH^−^, and Solv = EtOH, DMF or H_2_O) all possessing high porosity and surface area. The goal is the preparation of a hybrid system (MNPs@MIL) of nanosize dimensions (≤50 nm) which retains the magnetic properties of the iron core in order to allow magnetic separations and the loading capability of porous Fe-MILs. Moreover, toxicities of Fe_3_O_4_ nanoparticles and iron-based MIL are normally considered very low because of the biocompatibility of their component (iron and terephtalic acid). In particular, our previous results [13,38] suggested that magnetic nanoparticles with various organic coatings have negligible toxicity at the investigated concentrations of 20 μg/mL. The terephthalic ligand was considered to have a very low toxicity and its LD50 is more than 5000 mg/Kg by oral administration [39]. The Fe-MILs tollerability was reported to be above 1 mg/mL in vitro and more than 100 mg/Kg in vivo [40,41,42]. The nature and the morphology of the obtained hybrid nanomaterial was assessed through X-ray powder diffraction (XRD), X-ray photoelectron spectroscopy (XPS), Fourier transform infrared spectroscopy (FTIR), thermal gravimetric analysis (TGA), scanning electron microscopy (SEM) and transmission electron microscopy (TEM) characterizations and the load/release properties determined using a rhodamine dye as luminescent probe. Cell viability was evaluated by 3-(4,5-dimethylthiazol-2-yl)-2,5-diphenyltetrazolium bromide (MTT) tests. The cellular internalization and the high capability of MNP@MILs to deliver TMZ for the treatment of GBM were analyzed by optical microscopy.

## 2. Results

### 2.1. Synthesis and Characterization of Hybrid Nanoparticles

The adopted synthesis routes for the modification of the iron oxide magnetic nanoparticles (MNPs) are schematized in Figure 1.

The MNPs obtained by the co-precipitation method, are the magnetic core of the nanosystem but they are also an Fe^3+^ source for MIL growth. In particular, modified nanoparticles were obtained through two similar routes both using 2-aminoterephthalic acid as organic precursor.

In the first route Fe_3_O_4_ nanoparticles are the only source of the Fe^3+^ ions (Figure 1 route a) whilst in the second route (b) besides Fe_3_O_4_, another external source of Fe^3+^ ions (FeCl_3_) was added. The use of the iron nanoparticles themselves as Fe^3+^ ion source is a key point for the growth of the MIL structure on the MNPs. In addition, the synthetic protocols involve very low temperatures and short reaction times, thus preserving the structure and the properties of the iron oxide magnetic core.

The XRD diffraction patterns of the bare MNPs and the hybrid systems obtained after the functionalization of the magnetic Fe_3_O_4_ nanoparticles with iron-based MILs through the two routes are reported in Figure 2a. Patterns of MNPs show peaks at 2θ = 30.3°, 35.7° and 43.3° related to (220), (311) and (400) planes of crystalline Fe_3_O_4_ nanoparticles, confirming the presence of the magnetite phase [43]. After MNP functionalization with MILs through route (a) (MNPs@MIL[a]), no sharp reflections were detected, thus indicating that the formation of isolated fully crystallized MOFs was avoided [35,44,45]. A broad and very low signal was detected below 10° in the diffractogram of MNPs@MIL[a] and slightly more intense broad signals in the range 4–5° and 6–8° in the case of MNPs@MIL[b] obtained in the presence of FeCl_3_. These broad signals at low 2θ values are in the typical region observed for MIL phases and, in particular, for the MIL-101 phase [35,46]. The XRD patterns of MIL-101 powders is reported as a reference in the insert of Figure 2a. Note that compared to the powders, the signals were much broader and weaker because the MIL structures are grown on the MNPs and cannot crystallize into 3D grains that are typically greater that 200–400 nm [47].

The presence of MILs on the MNP nanoparticles is confirmed by FTIR. Figure 2b shows the FTIR spectra of the bare MNPs nanoparticles and of the MNPs@MIL systems. The most intense peak in the spectrum of Fe_3_O_4_ nanoparticles at 600 cm^−1^ is related to Fe−O stretching modes, typical of iron oxides.

The spectra of MNPs@MIL prove for both syntheses the MIL formation showing the typical vibrational modes of the aminoterephthalate ligands. In particular, the peak at 1250 cm^−1^ is typical of C-N symmetric and asymmetric stretching of the amino group. In addition, the signals at 1310–1420 cm^−1^ and 1510–1590 cm^−1^ are due to the COO^−^ symmetric and asymmetric stretching of the terephthalate ligands [46,47].

The chemical nature of the MIL layers coating the MNPs was investigated by XPS that is a surface sensitive technique. Table 1 shows the atomic composition of MNP surfaces before and after MIL growth. After MOF growth, the surface composition changes indicating the formation of MIL layers. In particular, surface analyses of MIL functionalized MNPs showed the increase of C concentration, as expected for the formation of a layer containing an organic ligand, and the decreased iron concentration due to the lower iron density in the MIL layers compared with the bare Fe_3_O_4_ oxides. Moreover, both MNPs@MIL spectra show a significant amount of N due to the amino groups of the aminoterephthalate ligands, whilst only a negligible concentration of N (about 0.2%) is present on bare MNPs due to adventitious contaminations. The main difference between MNPs@MIL[a] and MNPs@MIL[b] is the presence of chlorine. 

As expected, MNPs@MIL[b] showed the presence of Cl^−^ in the lattice according to the general formula [Fe_3_(O)X-(Solv)_2_(C_8_H_5_NO_4_)_3_] with Solv = EtOH or DMF and X = Cl^−^ because of the addition of FeCl_3_ during synthesis, whilst MNPs@MIL[a] did not show any chlorine suggesting X = OH^−^ in the general formula. 

Figure 3 compares N 1s (a) and Fe 2p (b) bands of bare MNPs, MNPs@MIL [a] and [b]. In both samples MNPs@MIL [a] and [b] N 1s peak consists of a broad signal at about 400.1 eV, typical of 2-aminoterphthalic ligand [47,48]. The Fe 2p signal (Figure 3b) showed a broad peak for all samples as observed for most of the iron compounds because of the presence of typical multiplet splittings [49]. The centroid of the signal is at 711.1 eV in the case of bare Fe_3_O_4_ due to the convolution of Fe^2+^ and Fe^3+^ ions [49]. After MIL growth the centroid is shifted to about 712.0 eV, likely due to the convolution of Fe^3+^ in the MIL structure (typically around 712.6 eV) and the Fe^3+^/Fe^2+^ ions of Fe_3_O_4_.

The evolution of the C 1s signal after the formation of the hybrid MIL structure is reported in Figure 4. Before MIL growth the C 1s signal on the bare MNPs is mainly due to the adventitious carbon [50,51] that is always observed in the XPS spectra and is typically constituted by a main peak at 285 eV and a small component at 286.2 eV due to oxidized carbons. A low intensity tail around 288–289 eV is also observed likely due to surface carbonate formation. After MIL growth, besides a relevant increase of the C 1s signal intensity (Table 1), the shape of C 1s is changed. In particular, for both MNPs@MIL [a] and [b] samples, the band (Figure 4) consists mainly of three components at 285, 286.5 and 288.6 eV. The most intense peak at 285 eV is attributable to aliphatic and aromatic hydrocarbon atoms and to the “adventitious” carbon [50,51]. The component centered at 286.5 eV is due to C-N atoms of the amino terephthalic moieties and to C-O groups of the ethanol, used as a solvent and probably adsorbed in MIL cavities. The peak at 288.6 eV is typical of carboxylate (-COO^−^) groups [47]. In addition, a broad and weak signal is detected at 289.4 eV and attributable to unreacted carboxylic acid (–COOH) [52].

Figure 5a–c shows SEM morphologies of Fe_3_O_4_ magnetic nanoparticles before (Figure 5a) and after their functionalization with MIL frameworks through both synthetic routes [a] and [b] (Figure 5b,c). For all samples, SEM images show homogeneous powders with grains having similar morphology and sizes, without the presence of spurious crystals. At the nanometric level, TEM images (Figure 5d–f) showed some morphological differences between MNPs before (Figure 5d) and after (Figure 5e,f) MIL formation. MNPs@MIL [a] particles show a slightly rougher and porous structure, which is much more evident for MNPs@MIL [b] samples indicating the occurrence of MNP modifications correlated to the formation of the iron-based MILs. Particles size distributions estimated from SEM and TEM images through Gwyddion software [53] are centered at about 25 nm, 30 nm and 40 nm for bare MNPs, MNPs@MIL [a] and MNPs@MIL [b], respectively, with a full half width of about 10 nm in all cases.

Unfortunately, it was not possible to investigate the crystalline structure of these hybrid systems through selected area electron diffraction (SAED) because of the well-known lability of MOF structures under electron beam analysis [54].

The amount of organic matter present in the system was determined through TGA. The TGA curve of bare MNPs in the 25–500 °C range shows (Figure 6) a total weight loss below 250 °C of about 2.2%, likely caused by the evaporation of adsorbed solvent and water molecules. The TGA curves of MNPs@MIL[a] and [b] show the same behavior of bare Fe_3_O_4_ particles until about 250 °C. Above this temperature, a larger weight loss compared to bare MNPs is observed for both MNPs@MIL systems, due to the degradation of the organic ligands of the MILs. For MNPs@MIL[a] and [b] the total weight losses are 4.9% and 6.4%, respectively. The comparison of these values with that of bare MNPs indicates that the organic component is 2.7% of the total mass for MNPs@MIL[a] and 4.2% for MNPs@MIL[b]. Assuming the general formula [Fe_3_(O)X-(ethanol)_2_(C_8_H_5_NO_4_)_3_] with X = OH^−^ for MNPs@MIL[a] and X = Cl^−^ for MNPs@MIL[b], the estimated weight % of the MIL in the MNPs@MIL powders is 3.6% and 5.7% for route (a) and (b), respectively.

### 2.2. Load/Release Experiments

The capability of the MNPs@MIL hybrid system of loading guest molecules was evaluated using Rhodamine B (Rhod) as an optical probe. In particular, the loading capability of MNPs@MIL[a] and [b] was compared to that of bare MNPs in order to determine the enhancement of the loading capability due to the presence of a MIL-based porous system. Figure 7a displays the evolution of residual Rhod concentrations (C) as C/C_0_ (where C_0_ is the initial concentration = 6 × 10^−6^ M) versus loading time. The naked nanoparticles (black line) show negligible adsorption capability since the change of Rhod concentration cannot be observed during the whole adsorption time. MNPs@MIL[b] (blue line) exhibits an initial fast loading during the first 30 min followed by a continuous loading with a lower rate in the next 90 min. Conversely, the MNPs@MIL[a] loading (green line) is slower and slightly less efficient. The different behavior of the two materials can be explained by both the increased percentage of organic component that covers the nanoparticles and the higher degree of crystallinity of the MIL, as evidenced by the TGA and the XRD pattern. Figure 7b compares the release profile of MNPs@MIL[a] and MNPs@MIL[b] in water. As shown, the amount of Rhod released by MNPs@MIL[a] (green line) is 0.4 mg/g, which is about 40% of the loaded Rhod amount (C_a_ = 1.0 mg/g). Conversely, the MNP@MIL[b] system allows a more efficient release of about 1.0 mg/g, which is 60% of the loaded Rhod amount (C_a_ = 1.8 mg/g) and it is two and half higher than MNPs@MIL[a]. Therefore, both systems MNPs@MIL[a] and [b] showed improved loading capabilities compared to bare Fe_3_O_4_ MNPs, likely because of the presence of the porous MIL structure and, in particular, MNPs@MIL[b] showed the best performance in terms of amount of loaded and released materials. For this reason, the functional properties of MNP@MIL[b] were exploited for the in vitro treatment of glioblastoma cells using TMZ as antitumoral drug.

Figure 8 shows the loading rate of TMZ from bare MNPs and MNPs@MIL[b]. For bare MNPs it was found that no drug is loaded onto the nanoparticles. However, in the case of MNPs@MIL[b] the percentage of the loaded drug increased remarkably over a total period of 2 h. The loaded quantity over time is in line with the results obtained during tests with the optical probe and, after 120 min, 23% of the initial concentration of the drug (about 12 mg/g) was already loaded onto the surface of MNPs@MIL[b].

### 2.3. In Vitro Cellular Uptake and TMZ Delivery Studies

Human glioblastoma cells (A-172) were incubated both with MNPs@MIL[b] and MNPs to study the fate of the endocytosed nanoparticles at different time points (up to 72 h). The uptake was assessed using the PerkinElmer Operetta High-Content Imaging System. As shown in Figure 9, we found that the internalization of MNPs@MIL[b] both in the cytoplasm and in the nucleus is more evident than bare MNPs at all the investigated concentrations. This behavior suggests that MNPs@MIL[b] is capable of targeting brain tumors with a greater efficacy than MNP.

In order to determine whether TMZ, MNPs@MIL[b] and TMZ-loaded MNPs@MIL[b] affect viability of human glioblastoma cells, we analyzed their effects on A172 cells. The cell line was treated with different concentrations of MNPs@MIL[b] (5–20 μg/mL) and TMZ (0.125–0.5 µM) for 72 h, then the cell viability was assessed by MTT assay. The results are shown in Figure 10. All tested concentrations of MNPs@MIL[b] reduced cell viability in a dose-dependent manner only at 48 h, while DMSO treatment, used as vehicle, did not show significant differences compared with untreated cells (Control) (Figure 10). After 48 h, all TMZ concentrations slightly decreased cell viability. After 72 h only TMZ concentrations of 0.375 and 0.5 µM caused a detectable decrease of cell viability. When MNPs@MIL[b] were loaded with TMZ, the cytotoxicity in A172 cells was significantly augmented compared to simple drug solution and basic material. In particular, after 48 h of combined treatment, we observed that MNPs@MIL[b] (15 and 20 μg/mL) suppressed A172 viability to about 50%.

## 3. Discussion

In the present study, a novel hybrid nanosystem consisting of an Fe-based MIL grown on a magnetic Fe_3_O_4_ core was successfully synthesized. The growth of MIL on the nanoparticles was obtained by two similar routes both using 2-aminoterephthalic acid as organic precursor. In the first route, Fe_3_O_4_ nanoparticles are the only source of the Fe^3+^ ions, whilst in the second route, besides Fe_3_O_4_, an external source of Fe^3+^ ions (FeCl_3_) was added. Bulk and surface chemical characterizations indicated for both routes the formation of a poor crystalline iron-based MIL structure on the Fe_3_O_4_ NPs, which retained both their crystallographic structure and magnetic properties since they can be easily magnetically separated. SEM and TEM images did not show the presence of isolated MOF crystals but the nanoparticles showed a rough and porous structure likely due to MIL growth that was evaluated through TGA to be between 3.6% and 5.7% depending on the synthetic route. In particular, MNPs@MIL nanopowders obtained with route (b) possess a higher amount of MIL compared to MNPs@MIL[a]. The different synthetic route also affected the loading/release properties of this hybrid material. Both systems, MNPs@MIL[a] and [b], showed improved loading capabilities compared with bare Fe_3_O_4_ MNPs, likely because of the presence of the porous MIL structure, and MNPs@MIL[b] showed the best performance in terms of amount of loaded and released materials. Compared to previous reported composite materials, our combined system keeps nanometer dimensions (within the nanometer scale, i.e., ≤50 nm) after MIL growth. Nanodimension is a typical requirement of drug-delivery carrier. The functional properties of MNPs@MIL[b] were therefore exploited through in vitro drug delivery experiments using TMZ as the drug for the treatment of glioblastoma. Experiments indicated that this hybrid system is capable of penetrating the A172 cell line more efficiently than bare MNPs and the use of TMZ-loaded MIL@MNPs particles increases the drug efficiency compared to free TMZ. Finally, our findings show that the combination of MIL frameworks and magnetic nanoparticles represents a promising approach to develop novel drug delivery systems.

## 4. Materials and Methods

### 4.1. Materials

Iron (II) chloride tetrahydrate (FeCl_2_ × 4H_2_O), Iron (III) chloride hexahydrate (FeCl_3_ × 6H_2_O), ammonium hydroxide (NH_4_OH), 2-aminoterephthalic acid [(H_2_NC_6_H_3_-1,4-(CO_2_H)_2_], ethanol (CH_3_CH_2_OH), dimethylformamide (DMF) and Rhodamine B (Rhod) were purchased from Sigma-Aldrich (Milan, Italy) and used as received. TMZ solution in DMSO 51.51 mM was purchased from Abcam (Cambridge, UK) and stored at −18 °C. The water was of Milli-Q grade (18.2 MΩ cm) and was filtered through a 0.22 mm filter.

### 4.2. Synthesis of Magnetic Iron Nanoparticles (MNPs) and Functionalization with Iron-Based MILs

MNPs were obtained through a co-precipitation method according to the procedure described in the following lines. NH_4_OH (5 mL, 25%) was added to a water solution (50 mL) obtained dissolving Fe^3+^ and Fe^2+^ in a molar ratio of 1:2, under an N_2_ atmosphere and constant stirring. The reaction was kept at 80 °C for 30 min. The resulting suspension was cooled to room temperature and washed with ultrapure water. The synthetized magnetic nanoparticles (MNPs) were isolated from the solvent by magnetic decantation. MIL frameworks were grown through two similar routes both using Fe_3_O_4_ nanoparticles as the metal core and source of Fe^3+^ ions. In the first route (a) magnetic Fe_3_O_4_ nanoparticles (0.25 g) were dissolved in an ethanol or DMF solution (15 mL) of 2-aminoterephthalic acid (0.16 g). The reaction was kept under reflux for 4 h in an 80 °C oil bath. Functionalized nanoparticles (MNPs@MIL[a]) were separated by magnetic decantation and rinsed several times in ethanol and water. The second route (b) was similar to the previous one but with the addition of FeCl_3_ × 6H_2_O (0.08 g) to the ethanol solution.

### 4.3. Characterizations

XRD measurements were performed with an XRD Smartlab Rigaku diffractometer (Tokyo, Japan) in grazing incidence mode (0.5°) operating with a rotating anode of Cu Kα source radiation at 45 kV and 200 mA. XPS was carried out with a PHI 5600 multi-technique ESCA-Auger spectrometer (Chanhassen, MN, USA) using a standard Mg-Kα X-ray source with a photoelectron take-off angle of 45° (relative to the sample Surface). The XPS binding energy (B.E.) scale was calibrated on the C 1s peak of adventitious carbon at 285.0 eV. Transmission FT-IR measurements of samples in KBr pellets were recorded using a JASCO FTIR 4600LE spectrometer (Easton, MD, USA) in the spectral range 560–4000 cm^−1^ (resolution 4 cm^−1^). The SEM images were obtained using a field emission scanning electron microscope (FESEM) ZEISS VP 55 (Oberkochen, Germany). TEM analysis was carried out with a JEOL JEM 2010F (Akishima, Tokyo, Japan) working at 200 kV accelerating voltage. The dried powder was dispersed mechanically on an ultra-thin carbon coated lacey carbon grid. TGA was performed using a Mettler Toledo TGA (Columbus, OH, USA) with STARe software. All the curves were acquired in air in the temperature range of 25–500 °C at a heating rate of 10 °C min^−1^.

### 4.4. Drug Loading and Release Experiments

Before carrying out the drug loading experiments, preliminary tests were performed with an organic dye. A stock solution of Rhodamine B (6 × 10^−6^ M) was prepared by dissolving the dye in water. Loading tests were performed by dispersing MNPs@MIL[a] or MNPs@MIL[b] (3 mg) in the Rhodamine B solution (4 mL). The solution was kept under stirring in an orbital shaker and, periodically, a small batch of the sample (1 mL) was removed and assayed. The amount of loaded simulant was analyzed using a JASCO V-560 UV/vis spectrophotometer (Easton, MD, USA) equipped with a 1 cm path length cell at 230 nm. The nanoparticles were isolated from the solution by magnetic separation and placed in pure water to perform the release test. The suspension was again kept under stirring in an orbital shaker and, periodically, a small amount of sample (1 mL) was removed and assayed through UV/Vis spectrometry. As regards TMZ loading tests, experiments were carried out similarly to Rhodamine B experiments above mentioned using a 2 × 10^−4^ M TMZ solution in water. The TMZ solution was prepared by dissolving 39 μL of the TMZ solution in DMSO (51.51 mM) in 10 mL of water. The amount of DMSO in the loading solution is, therefore, about 0.4%. Unfortunately, it was not possible to study TMZ release because of its low absorption coefficient and its maximum position (372 nm) which overlaps with that of the free aminoterephthalic ligand present in MIL solutions [55]. The drug loading and release studies were performed in triplicate.

### 4.5. Cell Culture

Human glioblastoma cells (A172) were cultivated in Dulbecco’s modified Eagle’s medium (DMEM) supplemented with 10% fetal bovine serum (FBS), 100 IU/mL penicillin and 100 IU/mL streptomycin at 37 °C in a humidified atmosphere with 5% CO_2_. When the cells were approximately 80% confluent, they were detached by Trypsin/EDTA-4Na (0.05%/0.02% *w*/*v*) and used for experiments. The cells were purchased from ATCC Company (Milan, Italy).

### 4.6. Cell Viability Assay

A172 cells were seeded into 96-well tissue culture plates at a concentration of 2 × 10^5^ cells per well. Cells were incubated at 37 °C in a 5% CO humidified atmosphere and maintained in the presence and absence of different concentrations (5, 10, 15, 20, μg/mL) of MIL for 24 h, using DMSO 0.5% and 1% as vehicle treated groups. Three hours before the end of the treatment time, 20 μL of 0.5% 3-(4,5-dimethylthiazol-2-yl)-2,5-diphenyltetrazolium bromide (MTT) in phosphate-buffered saline (PBS) was added to each microwell. Then the supernatant was removed and replaced with 100 μL of DMSO to dissolve the formazan crystals produced. The amount of formazan is proportional to the number of viable cells present. The optical density value at λ = 570 nm was measured using a microplate spectrophotometer reader (Synergy HT, BioTek Instruments, Inc., Winooski, VT, USA). Cell viability was expressed in % values with respect to the control. All experiments were performed in triplicate.

### 4.7. High-Content Screening (HCS)

Cells were seeded in 96-well plates (Cell Carrier™-96; PerkinElmer #6005550) at a density of 2 × 10^3^ cells per well. Nuclei were stained with NucBlue solution (NucBlue™ Live cell Stain, Thermo-Fisher Scientific #R37605) for 15 min at Room Temperature (25 °C) following the manufacturer’s instructions (Thermo Fisher Scientific, Waltham, MA, USA). After cell labelling for the nuclei, samples were washed three times in PBS and treated with DMSO, used as vehicle, and different MNPs@MIL[b] and MNP concentrations (5, 10, 15, 20 μg/mL). Cells were imaged using the PerkinElmer Operetta High-Content Imaging System (# HH12000000). Plates were read under confocal conditions using the 63× long WD objective. A specific fluorescence type channel was used to acquire images of NucBlue (Ex: UV light, Em: 460 nm) for nuclei staining, shown in blue. All images were analyzed using Harmony high-content imaging and analysis software (PerkinElmer, Waltham, MA, USA). Initial segmentation of cells was carried out in the DAPI channel by identifying the blue-stained nuclei with an area >30 µm. Finally, the number of spots per Area of Cytoplasm and Nucleus was expressed as mean per well.

## Figures and Tables

**Figure 1 ijms-23-02874-f001:**
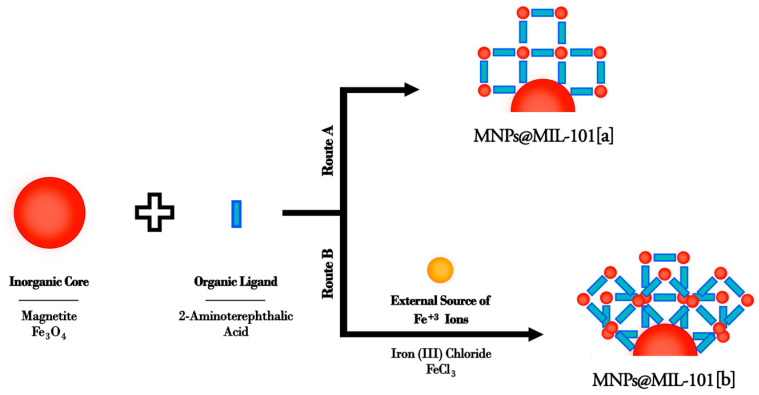
Generic scheme for the two synthetic routes proposed for the preparation of functionalized MNPs.

**Figure 2 ijms-23-02874-f002:**
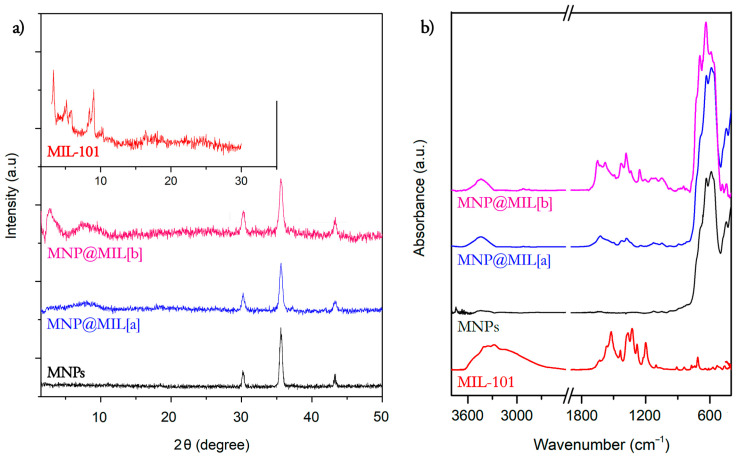
(**a**) XRD diffraction pattern of MNP nanoparticles (black line) and MNPs functionalized with MILs, obtained in the absence of FeCl_3_ (route a, blue line) and in the presence of FeCl_3_ (route b, magenta line). The insert is the XRD diffraction pattern of MIL-101. (**b**) FTIR spectra of MNPs nanoparticles (black line), MIL-101 powders (red line), and hybrid MNPs@MIL obtained through a (blue line) and b synthesis protocols (magenta line).

**Figure 3 ijms-23-02874-f003:**
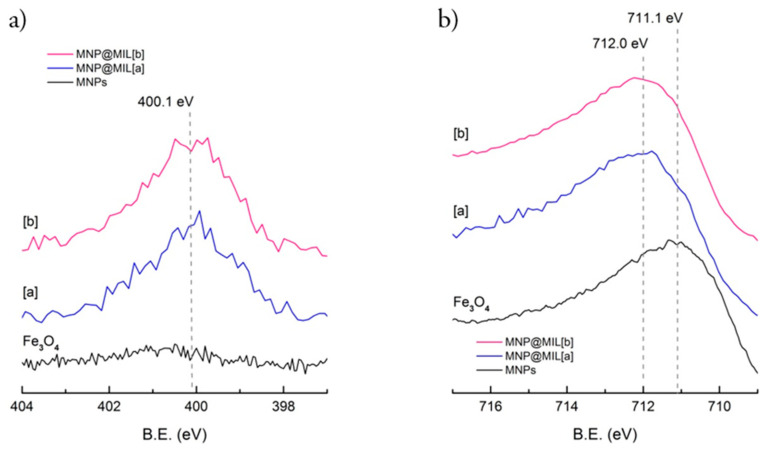
XPS spectra of N 1s (**a**), Fe 2p (**b**) of bare MNPs (**bottom**), MNPs@MIL[a] (**middle**) and MNPs@MIL[b] (**top**).

**Figure 4 ijms-23-02874-f004:**
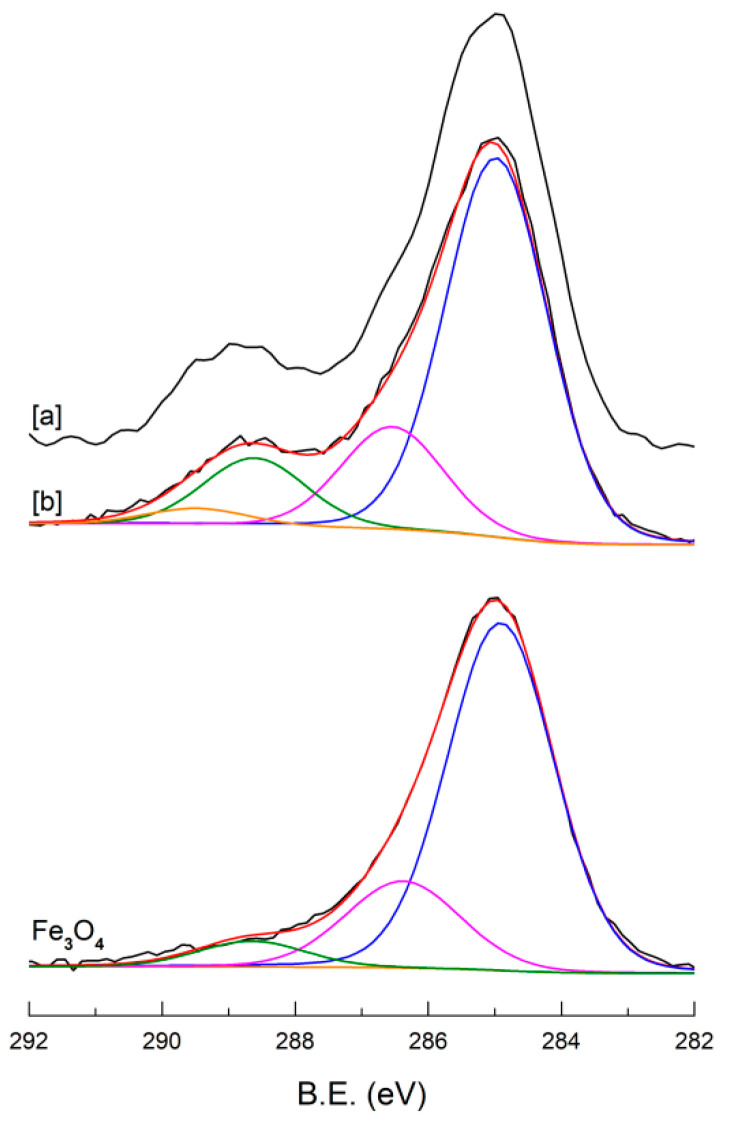
XPS spectra of C1s region of bare MNPs (**bottom**), MNPs@MIL[b] (**middle**) and MNPs@MIL[a] (**top**).

**Figure 5 ijms-23-02874-f005:**
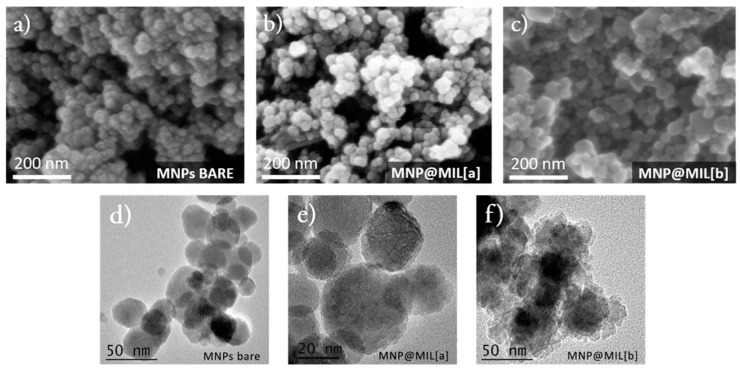
SEM morphology of (**a**) MNPs, (**b**) MNPs@MIL [a] and (**c**) MNPs@MIL [b]. TEM images of (**d**) bare MNPs, (**e**) hybrid MNPs@MIL[a] and (**f**) MNPs@MIL[b].

**Figure 6 ijms-23-02874-f006:**
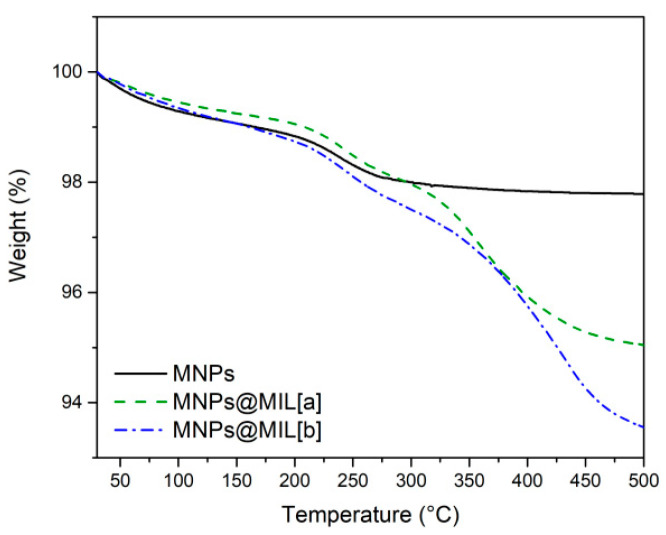
TGA plots of bare MNPs (black line), hybrid MNPs@MIL[a] (green line) and MNPs@MIL[b] (blue line).

**Figure 7 ijms-23-02874-f007:**
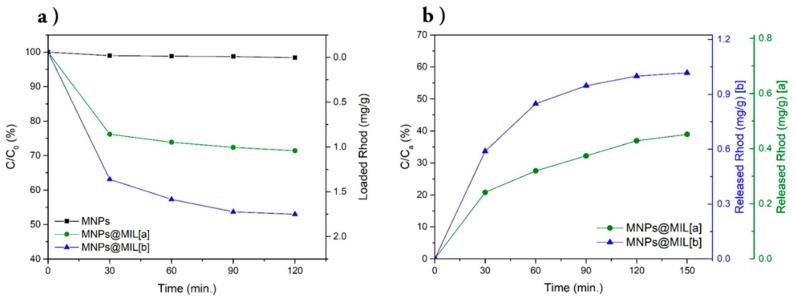
(**a**) Dependence of the residual Rhod concentrations (C) as C/C_0_ (where C_0_ is the initial concentration = 6 × 10^−6^ M) upon loading times using bare MNPs (black line), MNPs@MIL[a] (green line) and MNPs@MIL[b] (blue line); the right axis indicates the amount of loaded Rhod (mg/g); (**b**) Release plots of bare MNPs (black line), MNPs@MIL[a] (green line) and MNPs@MIL[b] (blue line). The amount of released probe is indicated as percentage of the loaded probe (C/C_a_ where C_a_ is the concentration (mg/g) of loaded Rhod); the right axes indicate the amount of released Rhod (mg/g).

**Figure 8 ijms-23-02874-f008:**
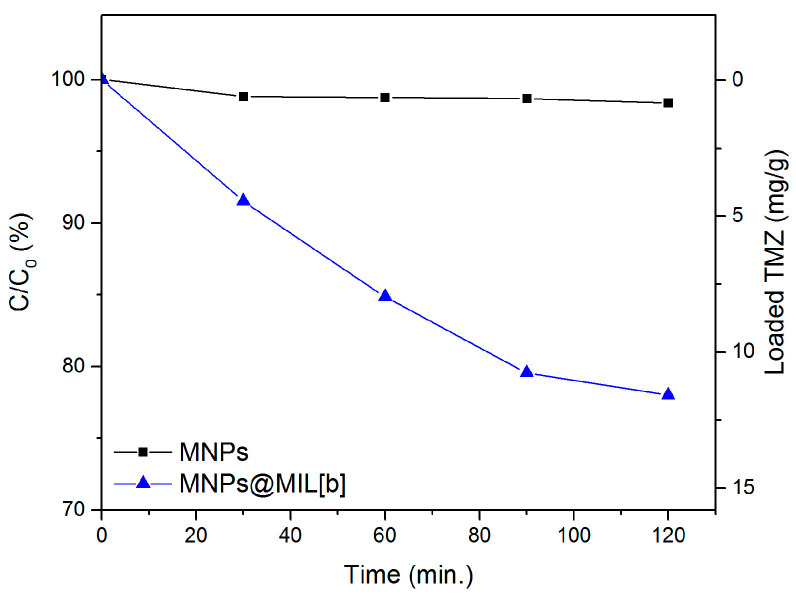
Dependence of the residual TMZ concentrations (C) as C/C_0_ (where C_0_ is the initial concentration = 2 × 10^−4^ M) upon loading times using bare MNPs (black line) and MNPs@MIL[b] (blue line); the right axis indicates the amount of loaded TMZ (mg/g).

**Figure 9 ijms-23-02874-f009:**
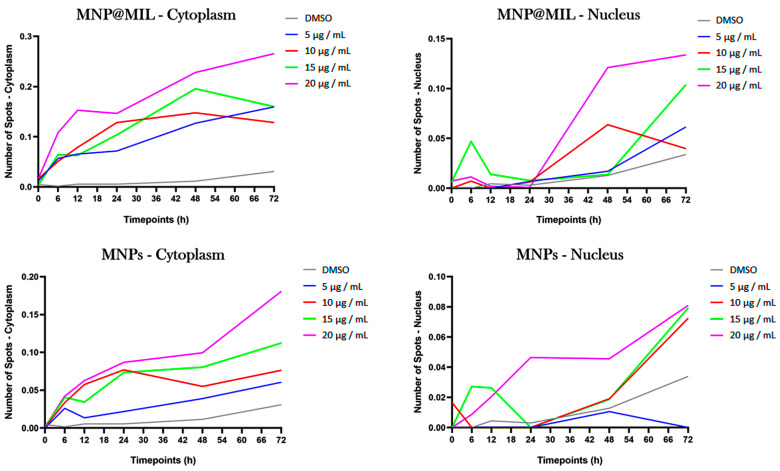
Internalization rate of MNPs@MIL[b] and MNPs in A-172 glioblastoma cells as measured by Operetta High-Content Imaging System. Data are reported as number of spots detected in cell cytoplasm and nucleus (time course 0–72 h). Data are presented as the average of four different replicates.

**Figure 10 ijms-23-02874-f010:**
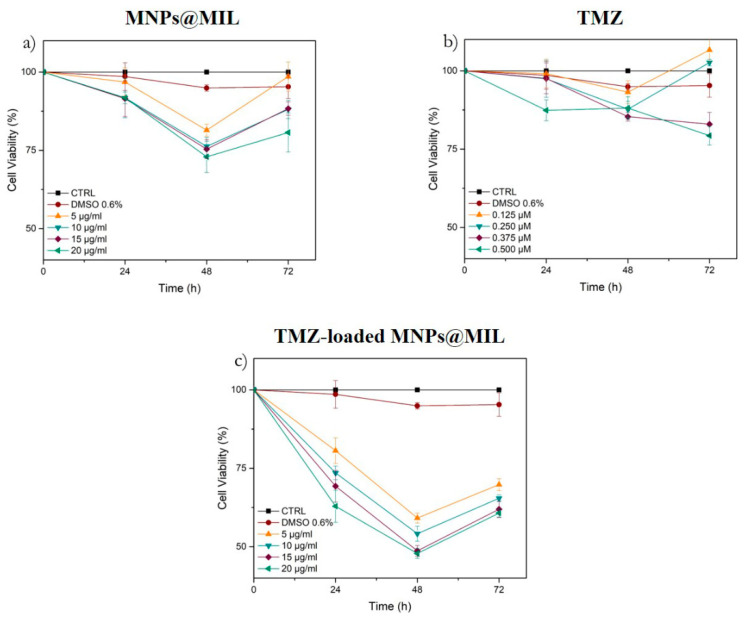
Cytotoxicity analysis of MNPs@MIL[b] (**a**), TMZ (**b**) and TMZ-loaded MNPs@MIL[b] (**c**) in human glioblastoma cells (A172) after 24 h, 48 h and 72 h. Data are presented as the average of three different replicates.

**Table 1 ijms-23-02874-t001:** XPS atomic concentration of MNPs, MNPs@MIL [a] and [b].

	XPS Atomic Concentrations				
	C 1s	O 1s	N 1s	Fe2p3	Cl 2p
MNPs	12.1	64.9	0.2	22.5	
MNPs@MIL[a]	27.2	53.8	2.0	17.0	-
MNPs@MIL[b]	35.3	51.3	2.1	10.7	0.6

## Data Availability

The data presented in this study are available and described in the article.

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
