# Peer review of "Synthesis of MIL-Modified Fe3O4 Magnetic Nanoparticles for Enhancing Uptake and Efficiency of Temozolomide in Glioblastoma Treatment"

_ijms, 2022, doi:10.3390/ijms23052874_

Round 1

Reviewer 1 Report

In this work the authors have developed a nanometric hybrid system consisting of a Fe3O4 magnetic nanoparticles modified through the growth of Fe-based Metal-organic frameworks of the MIL (Materials Institute Lavoiser). The obtained system possesses increased the loading capability due to the highly porous Fe-MIL and it was tested to load, carry and release temozolomide (TMZ) for the treatment of glioblastoma. The characterization of the hybrid system has been performed through several techniques.  This nanomaterial showed low toxicity and an increased adsorption capacity compared to bare Fe3O4 magnetic nanoparticles (MNPs). Although the experiments seem to be well performed and the study is very intereting, the authors should make several changes to make the paper suitable for publication in International Journal of Molecular Sciences.

Line 92: the authors mention the name of the method used for the synthesis of the MNPs (co-precipitation method). They should also indicate it in section 4.2.

Figure 1: the authors must formulate chemical compounds and ions well, that is, in the formulas they must use subscripts for the compounds and superscripts for the charges of the ions. For example, Fe3O4 insteaf of Fe3O4

Line 111: the sentence “…indicating that the formation of isolated fully crystalized MOFs was avoid.” must be supported by a reference.

Line 157: “…to about 712.0 likely…” appears and “…to about 712.0 eV likely…” should appear.

Lines 158-159: What is “The text continues here (Figure 2 and Table 2).”? I think that this sentence is a comment of the authors and it shouldn’t appear in the manuscript. In fact, Table 2 doesn’t appear in the paper.

Lines 174: the sentence “The most intense peak… to the “adventitious” carbon.” must be supported by a reference.

Lines 176-183: this paragraph should be rewritten besacuse it is confusing, they talk about SEM, after about XPS and finally about SEM again. It?s a mess.

SEM images: the authors state that the MNP@MIL obtained by the two routes show the same morphology. They should refute it by introducing in the manuscrpt the SEM images of both nanosystems.

TEM images: In general TEM images are usually accompained by the magification data used to obtain them. The authors should include this data in the revised manuscript.

In relation to the two microscopies used in this work, the authors talk about their particles having an approximate size of 50 nm (a fact to which they attach great importance for the application of their nanosystems). How did they measure the size? Have they used a computer program to measure it? How many nanoparticles did they use to give this size value? Which is the polidispersity in the size distribution? They should indicate all this in the revised manuscript.

Lines 227-231: the result of the amount of Rhod released by MNPs@MIL (a) shown in this paragraph does not agree with data shown in Figure 7b. Please, review these data ann rewrite this paragraph.

Line 242: “… plot of bare MNPS (black lines), MNPs@MIL[a]...” appear and “… plot of MNPs@MIL[a]...” should appear.

Lines 245-250: Why haven’t the authors studied the relese of TMZ?  I think this study is more interesting than the one of Rhod.

Line 264: the authors should introduce the following sentence after …their effets on A172 cells.”: The results are shown in Figure 10.

Lines 364-365: the authors say that the use an aqueous solution of TMZ, but they purchased a TMZ solution in DMSO, didn´t they? How did they prepare this aqueous solution?

Reviewer 2 Report

 The manuscript is well-written and has good structure.
It can be published.
However, I have some questions:
How about the toxicity of magnetic particles or the organic ligand?
How much can the living cells tolerate this metal organic framework?
What is the current method for drug delivery of TMZ?

Round 2

Reviewer 1 Report

The authors have taken into account all my suggestions as reviewer and have improved the work considerably. Under these conditions the paper is suitable for publication in International Journal of Molecular Sciences.